# SOL: Sampling-based Optimal Linear bounding of arbitrary scalar functions

**Yuriy Biktairov**    **Jyotirmoy Deshmukh**
University of Southern California
{biktairo, jdeshmuk}@usc.edu

## Abstract

Finding tight linear bounds for *activation* functions in neural networks is an essential part of several state of the art neural network robustness certification tools. An activation function is an arbitrary, nonlinear, scalar function $f : \mathbb{R}^d \to \mathbb{R}$. In the existing work on robustness certification, such bounds have been computed using human ingenuity for a handful of the most popular activation functions. While a number of heuristics have been proposed for bounding arbitrary functions, no analysis of the tightness optimality for general scalar functions has been offered yet, to the best of our knowledge. We fill this gap by formulating a concise optimality criterion for tightness of the approximation which allows us to build optimal bounds for any function convex in the region of interest $R$. For a more general class of functions Lipschitz-continuous in $R$ we propose a sampling-based approach (SOL) which, given an instance of the bounding problem, efficiently computes the tightest linear bounds within a given $\varepsilon > 0$ threshold. We leverage an adaptive sampling technique to iteratively build a set of sample points suitable for representing the target activation function. While the theoretical worst case time complexity of our approach is $O(\varepsilon^{-2d})$, it typically only takes $O(\log^\beta \frac{1}{\varepsilon})$ time for some $\beta \geq 1$ and is thus sufficiently fast in practice. We provide empirical evidence of SOL's practicality by incorporating it into a robustness certifier and observing that it produces similar or higher certification rates while taking as low as quarter of the time compared to the other methods.

## 1 Introduction

Neural networks (NNs) have become the pervasive machine learning model for various tasks including classification, function approximation, and generative modeling. In a typical classification task using supervised learning, the NN is given as input a high-dimensional data object (such as an image or a large piece of text) and its corresponding class label (from a finite set of labels). The NN classifier is deemed *robust* if for reasonable perturbations to a given input, the predicted label for the input does not change. It has been previously shown that NNs are susceptible to adversarial perturbations in inputs. These approaches take advantage of the non-robustness of a trained NN; for example, in image classification, by injecting a small amount of noise imperceptible to a human, the predicted label for an input image can be changed. In safety-critical applications where object detection and classification may be used as input for decision-making (e.g., in an autonomous mobile robot), such misclassifications can have safety implications. Similarly, when used in a function approximation task (e.g., for regression), it is important that small changes to the NN input do not lead to large changes in the predicted output. Checking the robustness of neural networks has thus emerged as an important problem in recent years [28, 3, 9, 1, 13, 23].

A number of different approaches have been developed for certifying the robustness of the neural networks with popular architectures. The majority of these approaches fall into one of three overar-

37th Conference on Neural Information Processing Systems (NeurIPS 2023).

ching categories: the algorithms using approximate calculations in order to speed the certification time up [25, 5, 33, 23, 13], the algorithms using exact calculations via some sort of constraint solver [10, 11, 29] and the algorithms taking advantage of both techniques [26, 24, 30, 32]. The most practical approaches which can handle the largest neural networks are usually built on top of the approximate techniques and rely heavily on being able to find tight linear bounds of non-linear scalar functions.

A lot of research has been dedicated to increasing the quality of linear bounds of the most popular activation functions such as ReLU[19], sigmoid, hyperbolic tangent and others[32, 25, 34, 36, 23, 13]. But not much has been done in the direction of efficiently bounding arbitrary functions. This led to the current situation where a number of well-performing novel activation functions like GeLU[8, 21] and Swish[22] and the neural networks containing them are effectively uncertifiable due to their linear bounding not being supported by the robustness certifiers.

To this end we introduce a novel approach for efficient optimal linear bounding of general scalar functions. The considered optimality is with respect to the conventional measure of tightness – the volume of the discrepancy region between the function and the linear bound. At the core of our approach is a simple idea of leveraging the Lipschitz continuity of the activation function in order to build a near-optimal bound using appropriately constructed sample of points from the region of interest. The major contributions of the presented paper are the following

- we analyze the problem of optimal linear bounding of scalar functions and derive a simple optimality criterion which allows us to optimally bound arbitrary functions convex or concave in the region of interest;

- we propose a novel optimal linear bounding approach called SOL which unlike any other known method efficiently produces linear bounds which are arbitrarily close to optimum for the general class of functions Lipschitz-continuous in the region of interest;

- we show that using SOL for the robustness certification is viable by benchmarking its performance on a synthetic dataset of bounding problems and by comparing it to the known alternatives on real neural network certification tasks.

## 2 Related work

To the best of our knowledge, there are only two approaches presented in the literature capable of producing linear bounds for arbitrary scalar functions. The first one is implemented inside the AutoLiRPA[35] framework for the robustness certification. The second is the LinSyn[20] approach.

AutoLiRPA supports bounding several general binary operations: addition, subtraction, multiplication and division. It is also capable of bounding compositions of these operations, which thus allows it to bound an arbitrary function decomposable into a sequence of some elementary operations. While this class of functions is indeed quite general and contains all the novel activation functions, the decompositional nature of the approach inevitably leads to producing bounds which are significantly suboptimal.

The authors of LinSyn, on the other hand, use an idea reminiscent of what we propose in this paper: they take a uniform sample of points from the region of interest, find the tightest bound for this finite set of points and then repeatedly adjust the obtained bound until its soundness is successfully certified by an SMT solver[4]. Unlike SOL, however, their approach does not provide any optimality guarantees for the generated bounds and heavily relies on the use of the SMT solver, which might limit the method's running time performance.

We provide an empirical robustness certification performance comparison between these two approaches and SOL in section 5.3 of this paper.

## 3 Optimal linear bounding

In this section we give a formal definition of the optimal linear bounding problem and derive the optimality criterion. We then use this criterion to give a recipe for the optimal linear bounding of functions which are convex or concave in the region of interest.

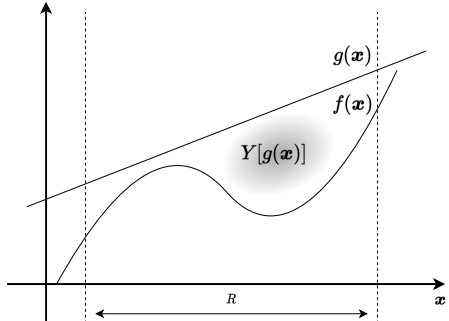

Figure 1: Linear bounding example.

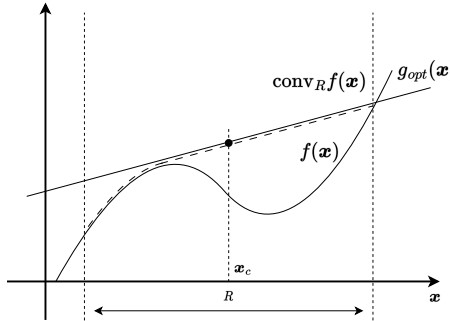

Figure 2: Optimal bounding criterion.

Without the loss of generality from now on we are only going to discuss the optimal upper bounding. The lower bounding problem can easily be reduced to the former by negating the function and then negating the values of the bound.

**Definition** (Optimal linear bounding problem). *Given a function $f : R \to \mathbb{R}$ defined in a convex bounded region $R \subset \mathbb{R}^d$, we aim to find such linear function $g(\boldsymbol{x}) = \boldsymbol{a}^\top \boldsymbol{x} + b$, that*

1. *$g(\boldsymbol{x})$ upper-bounds the original function: $g(\boldsymbol{x}) \geq f(\boldsymbol{x})$ for every $\boldsymbol{x} \in R$;*

2. *the volume $Y[g(\boldsymbol{x})] = \int_{\boldsymbol{x} \in R} |f(\boldsymbol{x}) - g(\boldsymbol{x})| d\boldsymbol{x}$ of the discrepancy region between the function graphs is as small as possible: $g_{opt}(\boldsymbol{x}) = \arg\min_{g(\boldsymbol{x})=\boldsymbol{a}^\top\boldsymbol{x}+b} Y[g(\boldsymbol{x})]$.*

The value of the discrepancy volume defined above is the conventionally accepted measure of tightness of the linear bound and is known to correlate well with the performance of robustness certification [36, 25, 13]. The notion is illustrated in the figure 1. We are going to call any linear function satisfying the two conditions an "optimal bound". Upon a detailed examination it can be noted that the discrepancy volume may be expressed in a more convenient form.

**Proposition 1.** *For any function $g(\boldsymbol{x})$ upper-bounding the target function $f(\boldsymbol{x})$ the equality $Y[g(\boldsymbol{x})] = g(\boldsymbol{x}_c) \cdot V + C$ holds, where $V = \int_{\boldsymbol{x} \in R} d\boldsymbol{x}$ is the volume of $R$, $\boldsymbol{x}_c = \frac{1}{V} \int_{\boldsymbol{x} \in R} \boldsymbol{x} d\boldsymbol{x}$ is the region's center of mass and $C$ is a constant independent of the function $g(\boldsymbol{x})$.*

*Proof.* Indeed, for any upper bound $g(\boldsymbol{x})$ the integrand in the definition of $Y[g(\boldsymbol{x})]$ can be simplified

$$\int_{\boldsymbol{x} \in R} |f(\boldsymbol{x}) - g(\boldsymbol{x})| d\boldsymbol{x} = \int_{\boldsymbol{x} \in R} (g(\boldsymbol{x}) - f(\boldsymbol{x})) \, d\boldsymbol{x} = \int_{\boldsymbol{x} \in R} g(\boldsymbol{x}) d\boldsymbol{x} + C,$$

where $C = -\int_{\boldsymbol{x} \in R} f(\boldsymbol{x}) d\boldsymbol{x}$ does not depend on $g(\boldsymbol{x})$. The straightforward integration

$$\int_{\boldsymbol{x} \in R} g(\boldsymbol{x}) d\boldsymbol{x} = \int_{\boldsymbol{x} \in R} (\boldsymbol{a}^\top \boldsymbol{x} + b) d\boldsymbol{x} = \boldsymbol{a}^\top \boldsymbol{x}_c V + bV = g(\boldsymbol{x}_c) \cdot V$$

then yields the desired expression. $\square$

Since $R$ is convex, $\boldsymbol{x}_c$ always lies in $R$. Then, $g(\boldsymbol{x})$ being an upper bound for the target function gives us a lower bound on the discrepancy volume $Y[g(\boldsymbol{x})] = V \cdot g(\boldsymbol{x}_c) + C \geq V \cdot f(\boldsymbol{x}_c) + C$. For a concave target function $f(\boldsymbol{x})$ this inequality concludes the problem of optimal linear bounding. Indeed, for a concave target function there always exists such an upper bound $g^*(\boldsymbol{x}_c)$ that $g^*(\boldsymbol{x}_c) = f(\boldsymbol{x}_c)$. This upper bound is then optimal due to the inequality.

Now, let $\text{conv}_R f(\boldsymbol{x})$ be the upper convex hull of the function $f(\boldsymbol{x})$ in the region $R$ – a function of $\boldsymbol{x}$ representing the upper boundary of the convex hull of $f$'s graph as is illustrated in figure 2 by the dashed line. The following then holds for arbitrary, possibly non-concave, functions $f(\boldsymbol{x})$.

**Proposition 2.** *A linear function $g(\boldsymbol{x})$ upper-bounds the target function $f(\boldsymbol{x})$ in the region $R$ iff it upper-bounds the convex hull $\text{conv}_R f(\boldsymbol{x})$ of the target function in the region $R$.*

*Proof.* If $g(\boldsymbol{x})$ does not upper-bound the target function $f(\boldsymbol{x})$, then it does not upper-bound its convex hull either since $\mathrm{conv}_R\, f(\boldsymbol{x}) \geq f(\boldsymbol{x})$ for every $\boldsymbol{x} \in R$. Conversely, if $g(\boldsymbol{x})$ does not upper-bound the convex hull there is at least one point $\boldsymbol{x}_0$ such that $f(\boldsymbol{x}_0) > g(\boldsymbol{x}_0)$. Otherwise the whole convex hull would lie below the linear function. $\qquad\square$

This proposition allows us to reduce the problem of bounding an arbitrary function $f(\boldsymbol{x})$ to bounding a concave function $\mathrm{conv}_R\, f(\boldsymbol{x})$. This reduction combined with the reasoning suggested for the concave target functions above gives us the following concise optimality criterion for the linear upper-bounding problem.

**Theorem 1.** *A linear upper bound $g(\boldsymbol{x})$ of a function $f(\boldsymbol{x})$ is optimal iff $g(\boldsymbol{x}_c) = \mathrm{conv}_R\, f(\boldsymbol{x}_c)$.*

Figure 2 illustrates the criterion showing a portion of $\mathrm{conv}_R\, f(\boldsymbol{x}_c)$ distinct from $f(\boldsymbol{x})$ with a dashed line. Theorem 1 immediately provides us with two efficient optimal bounding algorithms for concave and convex functions.

For concave and differentiable $f(\boldsymbol{x})$ the convex hull $\mathrm{conv}_R\, f(\boldsymbol{x})$ coincides with the function itself. Finding an optimal upper bound is then simply a matter of finding a hyperplane tangent to $f(\boldsymbol{x})$ at the center of mass $\boldsymbol{x}_c$. This only takes $O(d)$ operations if we have the access to the function's gradient.

For convex $f(\boldsymbol{x})$ and $R$ being a polytope one can show that only the values $f(\boldsymbol{x}_i)$ of the function in the vertices $\boldsymbol{x}_i$ of $R$ matter. The problem of upper-bounding then essentially becomes discrete – a linear program with $d$ variables and $|\{\boldsymbol{x}_i\}|$ constraints. In the typical case of a cubic $R$ this means having $O(2^d)$ constraints which can still be viable for reasonably high-dimensional functions. The process of solving this kind of linear programs efficiently is discussed in details and analyzed experimentally in sections 4.1 and 5.1 of this paper.

The two algorithms combined form a complete optimal upper- and lower-bounding suite to be used with arbitrary convex or concave differentiable activation functions. Optimal bounding of both sides of the function can be done exactly in this case with no need for approximations. Among the suitable popular activation functions are: softplus[37], ELU[2] and SELU[12] with appropriate choices of parameters. The approach can easily be generalized to piecewise-differentiable convex or concave functions as well. This extends the applicability to such functions as ReLU[19], leaky ReLU[16], SELU with a wider set of suitable parameters and the max pooling function (although not an activation function, also frequently used in neural networks).

# 4 SOL

In order to solve the problem of optimal linear bounding for the functions which are neither convex nor concave in $R$ we introduce the SOL approach: Sampling-based Optimal Linear bounding algorithm. In contrast to the approaches discussed above where an exactly optimal solution could be found in finite time, here we only aim at synthesizing an upper bound within a certain required tolerance target $\varepsilon$ of the optimum in terms of the discrepancy volume. In return, we get an algorithm which efficiently upper-bounds a very general class of activation functions – functions Lipschitz-continuous in $R$. Some of the popular non-convex and non-concave Lipschitz-continuous activation functions are: Sigmoid, Tanh, GeLU [8], Swish [22], Log Log[6], Mish [18], ELU and SeLU with $\alpha > 1$.

The key idea of SOL is to perform multiple iterations alternating between constructing an appropriate sample of points $S \subset R$ and solving the discrete version of the optimal bounding problem for the constructed sample until properly adjusted solution of the discrete problem meets the required accuracy target. During these iterations the generated sample progressively adapts to the target function increasing its density near the points where the optimal bound "touches" the function which allows for higher accuracy. Figure 3 illustrates the adaptive nature of these iterations by showing sample $S$ in the first 3 iterations of a SOL run.

## 4.1 Discrete problem

We define the discrete linear upper-bounding problem mentioned above in the following way.

**Definition** (Discrete optimal linear bounding problem). *Given a set of points $S = \{(\boldsymbol{x}_i, y_i)\}$, where $\boldsymbol{x}_i \in \mathbb{R}^d$ and $y_i \in \mathbb{R}$, and a center point $\boldsymbol{x}_c \in \mathbb{R}^d$, we aim to find such linear function $g(\boldsymbol{x}) = \boldsymbol{a}^\top \boldsymbol{x} + b$, that*

1. $g(\boldsymbol{x})$ *upper-bounds the sample points:* $g(\boldsymbol{x}_i) \geq y_i$ *for every* $(\boldsymbol{x}_i, y_i) \in S$;

2. *the value of the bound at* $\boldsymbol{x}_c$ *is as small as possible:* $g_{opt}(\boldsymbol{x}) = \arg\min_{g(\boldsymbol{x}) = \boldsymbol{a}^\top \boldsymbol{x} + b} g(\boldsymbol{x}_c)$.

Solving an instance of this problem with appropriately set $\boldsymbol{x}_c$ can be seen as relaxing an instance of the continuous optimal bounding problem to only contain a finite subset of the constraints while maintaining the original objective.

Clearly, the discrete problem is an instance of the linear programming (LP) optimization problem. It can be solved quite efficiently with one of many known approaches. We benchmark several popular general-purpose LP solvers as well as our implementation of two specific algorithms in the section 5.1 of this paper. It is worth noting that when the number of variables is bounded one can solve an LP in time linear in the number of constraints[17]. For our particular application it means that an instance of the discrete optimal linear bounding can be solved in $O(|S|)$ for a fixed $d$, which makes it as easy asymptotically as simply drawing the sample.

## 4.2 Soundness conditions

Relaxing the constraints of an optimization problem generally leads to a solution which is not feasible with respect to the original constraints. In case of SOL we hope to reason about the soundness of the bound in a continuous region based on the values of the target function in a finite set of points. One way to enable such reasoning is to limit the analysis to functions conforming to a certain smoothness criterion.

If the target function is "smooth enough" then given a subregion $U \subseteq R$, a point $\boldsymbol{x}^*$ within it and slope $\boldsymbol{a}$ of the solution to the discrete problem $g_S(\boldsymbol{x})$ there exists a certain threshold $\Delta(U, \boldsymbol{a}, f) > 0$ such that a big enough gap $g_S(\boldsymbol{x}^*) - f(\boldsymbol{x}^*) > \Delta$ implies $g_S(\boldsymbol{x}) > f(\boldsymbol{x})$ for every $\boldsymbol{x} \in U$. Having such a threshold allows us to obtain a sound upper bound in $U$ by appropriately shifting $g_S(\boldsymbol{x})$ up if needed. We analyze two different classes of target functions and show how the value of the threshold $\Delta$ depends on the parameters of the region and the smoothness of the target function.

### 4.2.1 $f$ is Lipschitz continuous in $R$

Let $L_1 > 0$ be an upper bound on the Lipschitz constant of the function $f$, meaning that

$$|f(\boldsymbol{x}_1) - f(\boldsymbol{x}_2)| \leq L_1 |\boldsymbol{x}_1 - \boldsymbol{x}_2|$$

for $\boldsymbol{x}_1, \boldsymbol{x}_2 \in R$. Let $g(\boldsymbol{x}) = \boldsymbol{a}^\top \boldsymbol{x} + b$ be the solution of the discrete problem associated with a particular sample of points $S \subset R$ containing $\boldsymbol{x}^*$. It can be shown that $|\boldsymbol{a}| \leq L_1 \cdot G$, where $G$ is a geometric factor determined by the shape of the region $R$ and the structure of the initial sample $S$. Importantly, in one-dimensional case (e.g., all traditional activation functions) the factor $G$ is exactly 1 for any region and any reasonable sample $S$. For an arbitrary point $\boldsymbol{x} \in U$ the following inequalities then hold

$$g(\boldsymbol{x}) - f(\boldsymbol{x}) \geq g(\boldsymbol{x}^*) - f(\boldsymbol{x}^*) - |\boldsymbol{a}||\boldsymbol{x} - \boldsymbol{x}^*| - L_1|\boldsymbol{x} - \boldsymbol{x}^*| \geq g(\boldsymbol{x}^*) - f(\boldsymbol{x}^*) - (1 + G)L_1\delta,$$

where $\delta = \sup_{\boldsymbol{x} \in U} |\boldsymbol{x}^* - \boldsymbol{x}|$ is the "width" of the region. Hence, as long as

$$g(\boldsymbol{x}^*) - f(\boldsymbol{x}^*) \geq \Delta = (1 + G)L_1\delta$$

the bound is sound for the whole subregion $U$.

The connection between $|\boldsymbol{a}|$ and $L_1$ originates from the following inequality which is based on $g(\boldsymbol{x})$ being a discrete upper bound for the $L_1$-continuous function $f(\boldsymbol{x})$

$$|\boldsymbol{a}| \leq L_1 \left( \inf_{\{\boldsymbol{x}_i\}, \boldsymbol{v}} \sup_{\boldsymbol{x} \in S, i} \boldsymbol{v} \cdot \frac{\boldsymbol{x} - \boldsymbol{x}_i}{|\boldsymbol{x} - \boldsymbol{x}_i|} \right)^{-1} = G(R, S)L_1,$$

where the infimum is taken over all finite subsets $\{\boldsymbol{x}_i\} \subset R$ such that the center $\boldsymbol{x}_c$ lies in the subset's convex hull and all unitary vectors $\boldsymbol{v}$. The factor $G(R, S)$ depends on the geometry of the target region $R$ and on the structure and the density of the sample $S$. Note that this factor decreases monotonically when we add new points to the sample, reaching a certain limit $G(R)$ when $S = R$.

**Algorithm 1** Adaptive SOL

---

$\delta \leftarrow \delta_0$
$\{U_i\} \leftarrow$ uniform partition into cubes of size $\delta$
$\boldsymbol{x}_c \leftarrow$ center of mass of $R$
$K \leftarrow \emptyset$
**do**
    $\{U_i\} \leftarrow \text{SPLITCUBES}(\{U_i\}, K)$
    $\boldsymbol{a}, b \leftarrow \text{SOLVEDISCRETE}(\{(\boldsymbol{x}_i, f(\boldsymbol{x}_i))\}, \boldsymbol{x}_c)$
    $\{\Delta_i\} \leftarrow \text{GETGAPTHRESHODLS}(\{U_i\}, \{\boldsymbol{x}_i\}, \boldsymbol{a}, b)$
    $K \leftarrow \{i | \Delta_i - (g_S(\boldsymbol{x}_i) - f(\boldsymbol{x}_i)) > \frac{\varepsilon}{V}\}$
**while** $K \neq \emptyset$
$b = b + \frac{\varepsilon}{V}$
**return** $\boldsymbol{a}, b$

---

### 4.2.2 $\nabla f$ is Lipschitz continuous in $R$

Let $L_2 > 0$ be an upper bound on the Lipschitz constant of the gradient $\nabla f$ of the target function. Given a bounded $R$ this immediately implies the Lipschitzness of the target function $f$ itself. Let $L_1$ be an upper bound on the Lipschitz constant of $f$.

The functions $f(\boldsymbol{x})$ of this type behave "approximately linearly" near the sample point $\boldsymbol{x}^*$. If we denote the linearized version of $f(\boldsymbol{x})$ as $f_l(\boldsymbol{x}) = f(\boldsymbol{x}^*) + \nabla f(\boldsymbol{x}^*) \cdot (\boldsymbol{x} - \boldsymbol{x}^*)$, the inequality $|f(\boldsymbol{x}) - f_l(\boldsymbol{x})| \leq \frac{1}{2} L_2 |\boldsymbol{x} - \boldsymbol{x}^*|^2$ holds for an arbitrary point $\boldsymbol{x}$.

As a consequence, the following inequality holds for the discrete bound $g(\boldsymbol{x})$

$$g(\boldsymbol{x}) - f(\boldsymbol{x}) \geq g(\boldsymbol{x}^*) - f(\boldsymbol{x}^*) - |\nabla f(\boldsymbol{x}^*) - \nabla g(\boldsymbol{x}^*)|\delta + \frac{1}{2} L_2 \delta^2.$$

Combined with the previous condition which is still applicable in this setting the inequality gives us the following expression for the gap threshold

$$\Delta = \min((1 + G)L_1\delta, \ |\nabla f(\boldsymbol{x}^*) - \nabla g(\boldsymbol{x}^*)|\delta + \frac{1}{2} L_2 \delta^2)$$

Intuitively, the introduction of this additional condition matters the most near the "touching points" between the function and the optimal linear bound. Around these points the gradients in the expression are going to be close to each other, so the second term in the minimization will allow for a significantly smaller gaps than those of the first term.

### 4.3 The algorithm

Now we have all the parts necessary to define the SOL algorithm as well as its simplified version.

At all times during the runtime of the algorithm we are going to maintain the partition of the original region into a collection of cubes $U_i = \left\{\boldsymbol{x} \in \mathbb{R}^d \middle| \bigwedge_{j=1}^d x_i^j - l_i \leq x^j \leq x_i^j + l_i\right\}$ with side lengths of $2l_i$ such that $R \subseteq \bigcup_i U_i$. The sample $S = \{\boldsymbol{x}_i\}$ we are going to use for the formulation of the discrete problem at a particular iteration is going to consist of the center points of these cubes.

Given the solution $g_S(\boldsymbol{x})$ of the discrete problem we are going to calculate the gap thresholds $\Delta_i$ associated with all $(U_i, \boldsymbol{x}_i)$ pairs according to one of the soundness conditions. The shift of value $\eta = \max_i [\Delta_i - (g_S(\boldsymbol{x}_i) - f(\boldsymbol{x}_i))]$ can then be applied to the discrete bound to obtain a bound sound in the entire target region $R$: $g_S^*(\boldsymbol{x}) = g_S(\boldsymbol{x}) + \eta$.

Importantly, since $g_S(\boldsymbol{x})$ is a solution of the relaxed problem its discrepancy is a lower bound for the optimal discrepancy. On the other hand, $g_S^*(\boldsymbol{x})$ is a feasible bound for the continuous optimization problem, hence its discrepancy is an upper bound for the optimal discrepancy. Combining these two observations with the proposition 1 we can conclude that $g_S^*(\boldsymbol{x})$ is within $\eta \cdot V$ of the optimum discrepancy volume-wise. The value of the shift can be directly controlled by varying the density of the cubes in the partition since $\eta \leq \max_i \Delta_i \leq (1+G)L_1 \max_i \delta_i$, where $\delta_i = \sup_{\boldsymbol{x} \in U_i} |\boldsymbol{x}_i - \boldsymbol{x}| = l_i \sqrt{d}$ is the size of cube $U_i$.

This observation inspires the uniform version of SOL. It only performs a single iteration with a uniform partition of $R$ into equally-sized cubes of size $\delta \leq \frac{\varepsilon}{(1+G)L_1 V}$, where $\varepsilon$ is the desired accuracy target. While being attractively simple this version turns out to be too slow in practice. Indeed, in a typical case we would only need such a high density of sample near the touching points of the optimal upper bound. Having sampled the whole region with a uniform density we end up with an intractably big discrete problem.

The solution to this issue is to start with a sparse regular lattice of cubes as a partition and gradually change it according to the values of $\Delta_i$ obtained after solving a discrete problem. Specifically, we would want to increase the density of cubes in such subregions $U_i$ that

$$\Delta_i - (g_S(\boldsymbol{x}_i) - f(\boldsymbol{x}_i)) > \frac{\varepsilon}{V}, \tag{1}$$

as these regions require too big of a shift in order to guarantee the soundness. In order to locally increase the density we split the target cube $U_i$ into $2^d$ evenly spread cubes of half the size of the original. An example of the split procedure can be seen in figure 3. This modification concludes the adaptive version of the SOL algorithm – algorithm 1.

### 4.4 On the complexity of SOL

Note that the cubes of size $\delta \leq \delta_c = \frac{\varepsilon}{(1+G)L_1 V}$ cannot violate the inequality (1) and, consequently, are never split. The maximum total number of points in $S$ achievable until the accuracy target is met is then defined by the critical cell size $\delta_c$ as $N_{tot} = O(\frac{V}{\delta_c^d}) = O(\varepsilon^{-d})$, where we only track the complexity's dependence on $\varepsilon$. Given that each iteration increases the sample size by at least 1 and solving the discrete problem as well as updating $S$ can be done in $O(|S|)$, we can bound the worst case complexity of SOL by $O(N_{tot}^2) = O(\varepsilon^{-2d})$.

While this runtime bound may seem excessively conservative, there exist linear bounding problem instances on which SOL is going to take similar $O(\varepsilon^{-\gamma})$ amount of time for certain values of $\gamma$. One particular example is bounding a linear function, where SOL would have to split all cubes down to the size of $\delta_c$ or $\delta'_c = \sqrt{\frac{2\varepsilon}{L_2 V}}$ depending on which soundness criterion is used. This example would give us the lower complexity bounds of $\Omega(\varepsilon^{-d})$ and $\Omega(\varepsilon^{-\frac{d}{2}})$ respectively just for the final iteration of SOL alone.

The situation is quite different, however, for many practical instances of the linear bounding problem. In a typical scenario the optimal bound $g_{opt}(\boldsymbol{x})$ only "touches" the target function $f(\boldsymbol{x})$ in $d$ separate points and $f(\boldsymbol{x})$ has nondegenerate Hessian in each of them. The high sample density of $\delta_c^{-d}$ or $(\delta'_c)^{-d}$ is then only necessary in the vicinity of these touching points. Taking the continuous limit one can show that verifying an $\varepsilon$-optimal linear bound in this case only requires $O(\log \frac{1}{\varepsilon})$ points in the sample $S$ when using the $L_2$ soundness condition. This logarithmic estimate of the number of points required for the verification is the primary reason we introduce the $L_2$ condition. Provided that the adaptive sampling procedure "finds" the optimal sample "fast enough" by splitting the cells in appropriate regions, one can hope to achieve the total running time of $O(\log^\beta \frac{1}{\varepsilon})$ for some $\beta \geq 1$ in arbitrary dimensionality $d$. We show that this is indeed the case in practice in section 5.2 where we measure the running time of SOL in a synthetic dataset of bounding problems involving popular activation functions.

## 5 Evaluation

In this section we study the performance of SOL empirically by measuring its runtime and the quality of the linear bounds on a synthetic dataset containing randomly generated instances of the bounding problem. We also show the practical applicability of SOL by incorporating it into AutoLiRPA and measuring its robustness certification performance on several image classification neural network architectures.

### 5.1 Benchmarking discrete problem solvers

Each run of SOL may involve dozens of invocations of the discrete problem solver. There are numerous different LP solvers that can be used for handling the discrete problem. Their performance

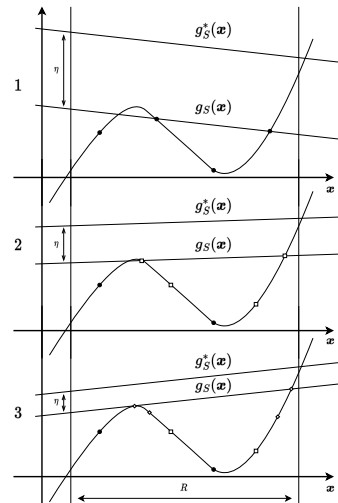

Figure 3: Adaptive sampling.

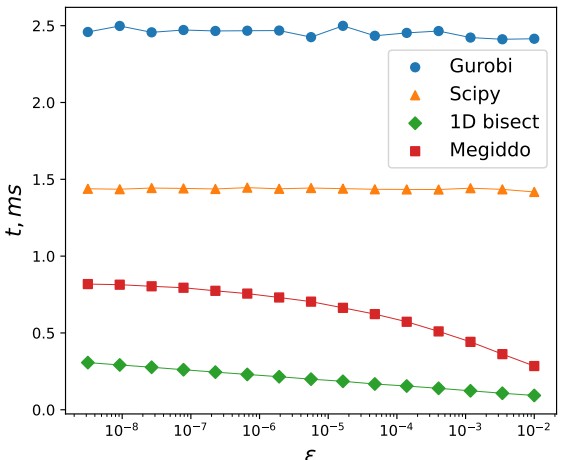

Figure 4: Runtime of LP solvers.

---

**Algorithm 2** 1D bisect algorithm

$u \leftarrow \max\{y_i\}$
$l \leftarrow \min\{y_i\}$
$a \leftarrow 0$
**while** $u - l > \varepsilon$ **do**
    $m \leftarrow \frac{l+u}{2}$
    $l_{max} \leftarrow \max\left\{ \left. \frac{m-y_i}{x_c-x_i} \right| x_i < x_c \right\}$
    $r_{max} \leftarrow \max\left\{ \left. \frac{y_i-m}{x_i-x_c} \right| x_i > x_c \right\}$
    **if** $r_{max} \leq l_{max}$ **then**
        $u \leftarrow m$
        $a \leftarrow \frac{l_{max}+r_{max}}{2}$
    **else**
        $l \leftarrow m$
    **end if**
**end while**
**return** $a, u$

---

is known to vary drastically between the different algorithms used at their core and the distributions of problems targeted. We benchmark 4 approaches on a synthetic dataset of the discrete problem instances to find the one most suitable for our applications.

The dataset was generated by sampling 2000 discrete problem instances for each of the following activation functions: GeLU, Log Log, Swish. We use these three activation functions as they have become quite popular in recent years[8, 21, 22, 27] but don't have optimal linear bounds handcrafted for them specifically. Each problem instance has the region's $R = [l, r]$ boundaries sampled uniformly from the $[-2, 2]$ interval and contains 500 points sampled uniformly from the region. This number of points was chosen for the experiments as corresponding to the typical final sample size of SOL in application to the robustness certification. Although the uniform distribution of points might not give an accurate representation of the samples encountered by SOL, the difference between the running times of the solvers is pronounced enough to ignore this inaccuracy.

The approaches we investigate include two well-known optimization libraries capable of solving LP: Gurobi[7] and SciPy[31]. The other two approaches are our implementation of Megiddo's linear-time algorithm[17] and a specific bisection procedure which takes advantage of the linear ordering of the sample points in the one-dimensional case. The idea behind the bisection procedure is that a particular $g(\boldsymbol{x}_c) = h$ guess on the objective value can be validated in linear time by checking whether

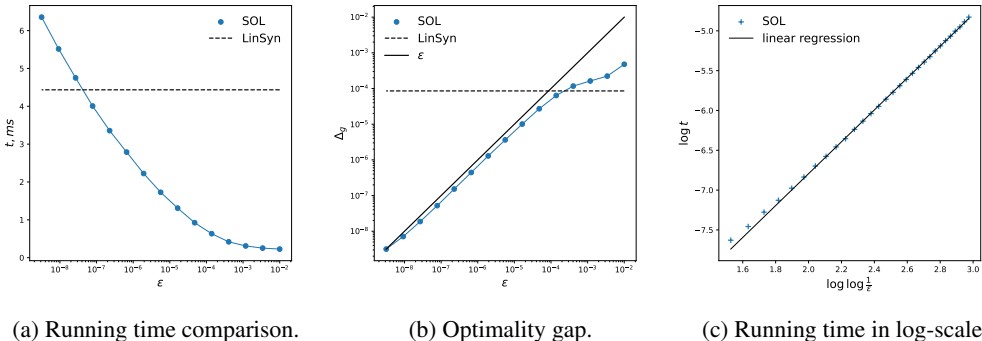

| (a) Running time comparison. | (b) Optimality gap. | (c) Running time in log-scale. |

Figure 5: SOL benchmarking results.

$(\boldsymbol{x}_c, h)$ lies inside the convex hull of the sample or not. A pseudocode implementing the procedure is shown in the algorithm 2. Note that this procedure is only applicable for $d = 1$.

The benchmarking results are shown in the figure 4. The plot shows the dependence of solver's average runtime across all the problems in the dataset on the targeted optimality tolerance $\varepsilon$. Interestingly, both general LP solvers show slower performance than the alternatives, which highlights the specific nature of linear programs encountered by SOL. Both of their running times seem almost independent of the optimality tolerance required. One possible reason would be the presence of significant overhead of setting up sophisticated optimization routines.

The results of the Megiddo's algorithm show expectedly saturating runtime dependence on $\varepsilon$ since it's supposed to find the exact solution in $O(|S|)$ time. The only dependence on $\varepsilon$ is due to the presence of an early stopping upon reaching the target accuracy. The bisection approach shows the best results with a significant margin along the entire bandwidth of $\varepsilon$ values. Although its dependence on $\varepsilon$ is asymptotically worse than that of the Megiddo's algorithm – $O(\log \frac{1}{\varepsilon})$ vs $O(1)$ – the constant factor of the latter appears to be too high for it to outperform the bisection procedure at practical values of optimality tolerance. Taking into consideration this result, we use the bisect algorithm as the discrete problem solver in all of the remaining evaluations.

## 5.2  Benchmarking SOL

In order to study the average runtime and the bounding performance of SOL on the linear bounding problem instances encountered in practice we generate another synthetic dataset. This time we sample 1000 instances of the problem for each of the same three functions. The region bounds for the problems are sampled uniformly from the $[-3, 3]$ interval.

We compare the SOL's performance on this benchmark to the performance of LinSyn[20] as their approach also aims to build bounds close to the optimal ones. It does not, however, give any guaranties on the optimality gap and does not have parameters controlling the accuracy target. Its metrics are thus independent of the optimality target $\varepsilon$.

Figures 5a and 5b show the average runtime and the average optimality gap respectively as functions of the optimality target $\varepsilon$. The optimality gap is estimated by taking the most accurate linear bound produced by SOL and shifting in down appropriately to obtain a tight underestimate of the optimal value of objective. Notably, there appears to be a wide range of accuracy targets $\varepsilon \in [10^{-7}, 10^{-3}]$ where SOL gives better average bounds while taking less time then LinSyn. For example, at $\varepsilon = 10^{-5}$ the SOL bounds are 10 times tighter than these of LinSyn while the runtime is 3 time shorter.

While figure 5a already hints on the form of the asymptotic dependency of the running time $t$ on the accuracy target $\varepsilon$, we make the analysis clearer by investigating the relation between $\log t$ and $\log \log \varepsilon$ instead. This dependence shown in figure 5c turns out to be quite close to linear with the slope of $a \approx 2.0$ estimated by the linear regression. The linear relation with this slope corresponds to the complexity of $O(\log^2 \frac{1}{\varepsilon})$ and agrees with our hypothesis on the typical running time of SOL.

Table 1: Certification performance comparison. The upper number is the fraction of properties certified, the lower is the estimate of the running time (with std $\leq 3\%$). Neural network architectures are denoted by the activation function used and their respective depths.

| Approach | Dataset | | | | | |
| --- | --- | --- | --- | --- | --- | --- |
| | MNIST [15] | | | CIFAR [14] | | |
| | GeLU 4l | Log Log 4l | Swish 4l | GeLU 5l | Log Log 5l | Swish 5l |
| AutoLiRPA [35] | 0.01 | 0.0 | 0.34 | 0.0 | 0.59 | 0.03 |
| | 230s | **23s** | **9s** | 700s | **50s** | **35s** |
| LinSyn [20] | 0.72 | 0.23 | **0.76** | **0.31** | **0.69** | 0.35 |
| | 430s | 450s | 410s | 580s | 360s | 550s |
| SOL ($\varepsilon = 10^{-3}$) | **0.73** | 0.23 | **0.76** | **0.31** | **0.69** | **0.37** |
| | **100s** | 96s | 100s | **150s** | 87s | 150s |
| SOL ($\varepsilon = 10^{-5}$) | **0.73** | **0.24** | **0.76** | **0.31** | **0.69** | **0.37** |
| | 140s | 140s | 130s | 190s | 95s | 170s |
| SOL ($\varepsilon = 10^{-7}$) | **0.73** | **0.24** | **0.76** | **0.31** | **0.69** | **0.37** |
| | 310s | 300s | 290s | 400s | 160s | 340s |

## 5.3 Robustness certification performance

Finally, we study the performance of SOL as a part of a robustness certification pipeline by using SOL as an activation function bounding subroutine in the AutoLiRPA[35] framework. We follow the protocol introduced in LinSyn and compare the robustness certification performance of our approach against that of LinSyn and that of AutoLiRPA with their default decompositional linear bounding approach. In these experiments we attempt the robustness certification of 6 image classification neural networks with convolutional architectures. Three of them were trained on MNIST[15] while the other three – on CIFAR[14].

Each certification task consists of proving that any $l_\infty$-bounded perturbation of a particular correctly-classified input image retains the same label prediction as the image itself. For each neural network we randomly sample 100 images from the test part of the dataset, filter out the misclassified ones and attempt to certify the remaining images. We use perturbation magnitude of 8/255 for networks trained on MNIST and 1/255 for CIFAR networks as is common in the literature. We have also conducted experiments with larger perturbations of 2/255 and 4/255 for the CIFAR models, but found the performance too weak for a proper analysis. Qualitatively the results for these larger perturbations were similar to the presented results.

The results are shown in table 1. Both LinSyn and SOL show much higher certification rates compared to AutoLiRPA since they aim at producing tight linear bounds while the default bounding procedure of AutoLiRPA systematically ends up with loose bounds.

The SOL variants show similar or slightly higher certification rates than LinSyn while taking as low as a quarter of the time. Lowering the accuracy target down from $10^{-5}$ noticeably increases the runtime of SOL without any improvement to the success rate of the certification. Notably, although not depicted in the table, the average sizes of the final bounds on the logit outputs of the networks only improve by a fraction of 1% upon changing $\varepsilon$ from $10^{-3}$ to $10^{-7}$ which is negligible compared to the improvement of x2-x5 from switching from AutoLiRPA to either LinSyn or SOL.

## 6 Conclusion

In this paper we have introduced SOL – a sampling-based approach for finding linear bounds arbitrarily close to optimum in terms of tightness for Lipschitz-continuous functions. We have shown experimentally that despite having rather pessimistic theoretical running time guarantees, it is quite fast in practice. As a part of a neural network robustness certification framework it increases the certification rates and takes significantly smaller time compared to the alternative approaches.

## Acknowledgments

The authors would like to thank the anonymous reviewers for their feedback. This work was supported by the National Science Foundation through the following grants: CAREER award (SHF-2048094), CNS-1932620, CNS-2039087, FMitF-1837131, CCF-SHF-1932620, funding by Toyota R&D and Siemens Corporate Research through the USC Center for Autonomy and AI, an Amazon Faculty Research Award, and the Airbus Institute for Engineering Research.

The authors would also like to acknowledge early discussions with Prof. Chao Wang and Dr. Brandon Paulsen that influenced this work.

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
