# OpenReview forum: "SOL: Sampling-based Optimal Linear bounding of arbitrary scalar functions"
_NeurIPS.cc/2023/Conference — NeurIPS 2023 poster_

### Official Review · Reviewer_8jE6 · 2023-07-06

**Soundness:** 3 good
**Presentation:** 3 good
**Contribution:** 3 good
**Rating:** 7
**Confidence:** 3

**Summary:**

This paper proposed a method to upper and lower bound a scalar function within a convex set using linear functions. The proposed method works by solving a sequence of discrete linear bounding problems with an increasing number of sample points.

**Strengths:**

1. The proposed method works for neural networks with general activation functions. Most robust verifier in the existing liturature can only handle popular activation functions such as ReLU, Sigmoid and Softmax.
2. This method achieve similar performance as the current state-of-the-art, LinSyn, while requires significantly less computational time.

**Weaknesses:**

1. The experimental section is a little bit weak. The proposed method does not seem to improve the "fraction of properties certified" compared to LinSyn in Table 1. From Table 1, it seems like the proposed method is simply a more efficient implementation of LinSyn.


**Questions:**

1. What kind of training method is used to train the models in Table 1? Are those models robustly trained to against $\ell_\infty$ attacks?
2. Table 1 suggests that lowering the $\epsilon$ down from $10^{-5}$ would only increase the runtime of SOL without gaining any improvement to the "fraction of properties certified". Is this empirical finding consistent among larger $\ell_\infty$ bound perturbations? Because the $\ell_\infty$ bound perturbation used in Table 1 is quite small, 8/255 for MNIST and 1/255 for CIFAR-10. Would SOL require smaller $\epsilon$ in order to achieve a similar "fraction of properties certified" as LinSyn?
3. It would be helpful to also include the upper bound on the "fraction of properties certified" in Table 1 in order to gauge how robust the models are. Such upper bound can be easily computed using projected gradient descent or algorithm for finding the attacks for the neural network.


**Limitations:**

The authors have adequately addressed the limitations.

---

> ### Author Rebuttal · Authors · 2023-08-10
>
> Thank you for your review and the feedback.
>
> [proposed method does not seem to improve the "fraction of properties certified" ...] We would like to point out that this can be easily explained by how LinSyn was tuned.
> While their method does not provide any tuning parameters for the user, it has a number of hardcoded parameters determining the trade-off between the running time and
> the accuracy. These parameters might have been chosen in a such a way as to make the bound tight enough for the certification rates to be close to saturation. This
> hypothesis seems quite realistic considering that the experimental setup we use is exactly the same.
>
> [training method] We train the models by optimizing a simple cross-entropy without implementing any robust training techniques. Comparison on the models trained
> specifically to be efficient at robustness certification would definitely be a useful extension.
>
> [scale of perturbation] We conducted additional experiments running SOL on CIFAR with perturbation radii 2/255, 4/255 and 8/255.
> Similarly to the results for 1/255 presented in the paper the certification rates for any other radius do not depend on whether
> the optimality target is set to 1e-7, 1e-5 or 1e-3. This means that for bigger radii the saturation still happens somewhere above 1e-3.
> The certification rates drop significantly with increased radius as can be seen here
> ```
>       gelu, loglog, swish
> 1/225 0.31  0.69    0.37
> 2/225 0.    0.38    0.02
> 4/225 0.    0.08    0.
> 8/225 0.    0.03    0.
> ```
>
> Supposedly, the drop is so steep due to the models themselves not being very robust to begin with.
> The running times also increase with increased perturbation scale. The change is more pronounced for the small eps = 1e-7
> ```
>       gelu, loglog, swish
> 1/225 400s  160s    340s
> 8/225 700s  312s    630s
> ```
> then for the larger eps=1e-3
> ```
>       gelu, loglog, swish
> 1/225 150s  87s     150s
> 8/225 179s  90s     165s
> ```
>
> We hope to get the LinSyn results for the increased radii soon, so that we can make the comparison complete.
> However, so far nothing indicates that the comparison might be qualitatively different from the 1/255 comparison presented in the paper.
>
> [optimization-based bound on the certification rates] Thank you for the suggestion! It might, indeed, give us new insights. We'll try to implement such bounds in a couple
> of days.

---

> > ### Author Response · Authors · 2023-08-14
> > **Increased perturbation radii results**
> >
> > We apologize for the delay. Here are the LinSyn results on CIFAR with increased perturbation radii.
> >
> > LinSyn certification rates:
> > ```
> >       gelu, loglog, swish
> > 1/225 0.31  0.69    0.35
> > 2/225 0.    0.38    0.02
> > 4/225 0.    0.08    0.
> > 8/225 0.    0.03    0.
> > ```
> > LinSyn runtimes:
> > ```
> >       gelu, loglog, swish
> > 1/225 582s  355s    546s
> > 2/225 576s  349s    562s
> > 4/225 586s  361s    563s
> > 8/225 599s  360s    574s
> > ```
> > The certification rates for SOL are indeed at least the same as those of LinSyn across the whole spectrum of perturbations.
> > Interestingly, the certification runtime does not increase much for LinSin between perturbations of 1/255 and 8/255.
> > However, it is still much higher than that of SOL with $\varepsilon = 10^{-3}$.
> >
> > To better analyze the saturation of certification rates as $\varepsilon \rightarrow 0$ we also evaluate SOL with $\varepsilon = 10^{-2}$ and achieve the following rates
> > ```
> >       gelu, loglog, swish
> > 1/225 0.25  0.62    0.32
> > 2/225 0.    0.33    0.
> > 4/225 0.    0.08    0.
> > 8/225 0.    0.03    0.
> > ```
> > The results suggest that for the perturbations of 1/255 and 2/255 the saturation occurs somewhere between $10^{-3}$ and $10^{-2}$, while for the perturbations of 4/255 and 8/255 the value of $\varepsilon = 10^{-2}$ is already saturated. Seemingly, the saturation $\varepsilon$ increases with perturbation scale which should be beneficial in practice.
> >
> > The LinSyn results also support this saturation dynamics hypothesis: the implicit optimality accuracy LinSyn has is not quite good enough for it to saturate the certification rates at 1/255 (SOL rate for the swish network is better), but is enough to saturate the rates for 2/255, 4/255 and 8/255.
> >
> > Finally, we would like to emphasize that the main purpose of the evaluation presented in the paper was to compare SOL to the two known function-agnostic bounding approaches.
> > The evaluation on a more diverse set of NNs trained using various techniques would surely enrich the analysis. However, since none of the approaches rely explicitly
> > on any specific model characteristic or the training algorithm, we argue that the presented results are decisive enough to be representative on their own. Especially
> > considering the following:
> > 1) we follow the experimental setup of LinSyn, which, supposedly, was chosen by the authors to reflect the performance of their approach as well as possible;
> > 2) LinSyn aims to produce bounds tight in the same $L_1$-distance sense as our optimal bounds, so it makes sense to attribute the difference in the robustness certification
> > performance to SOL being able to produce tighter bounds using less time (see fig. 7); this should stay true for any alternative experimental setup.

---

> > > ### Comment · Reviewer_8jE6 · 2023-08-16
> > >
> > > Thank you for the detailed response and the additional experimental results. I think the reason for the certification rate to drop so quickly is probably because the model is not robustly trained. It would be interesting to find out whether it is actually the case. Based on the current results, I think the proposed method should be strictly better than LinSyn as long as $\epsilon$ is not chosen to be too small. I have increased my score to 7.
> > >
> > > To better demonstrate the efficiency and efficacy of the proposed method. It would be really beneficial to include additional experiments using robustly trained models, and also include the upper bound on the certification rate: defined to be the fraction of inputs that the attack algorithms fail to find an attack. Maybe the runtime would reduce and the certification rate would not drop rapidly for SOL in larger radii with the robustly trained model.

---

### Official Review · Reviewer_GjZS · 2023-07-12

**Soundness:** 3 good
**Presentation:** 3 good
**Contribution:** 2 fair
**Rating:** 6
**Confidence:** 4

**Summary:**

This paper describes an approach for finding a linear upper bound of scalar functions that are Lipschitz. To find a bound that approximately optimizes discrepancy with the target function, it samples points to construct LP instances whose solutions bound the corresponding "discrete" bounding problem, and continues until finding a bound that matches an optimality target. This technique can be applied as a primitive in neural network verification routines, and the evaluation shows that when it is used in place of prior work, it leads to faster verification times with no penalty on accuracy for MNIST and CIFAR10 models.

**Strengths:**

This work addresses a specific problem with well-known applications, introducing a new technique that improves measurably on prior work. Notably, while the performance of this approach when used in robustness verification is quite a bit better than the recent best-in-class techniques, it does not seem to impose any additional restrictions that would limit its applicability to networks with different architectures or activation functions, and it does not seem to degrade the precision of the verifier.

Unlike the most closely related work, this approach provides a parameter that can be tuned (i.e., the approximation threshold) to trade performance for precision. While this isn't explored too much in the paper, this flexibility could potentially be used by verifiers to adaptively scale to larger problems by offering progressively weaker guarantees.

The writing is clear and understandable, although the main contribution takes some time to get through. The writing might be improved with a  more focused exposition of the algorithm. Currently, some of the intuition is given in section 3, some at the beginning of section 4, and the full algorithm is finally presented in 4.3. Soundness conditions and a brief diversion into LP are discussed in between these sections, which was distracting.

**Weaknesses:**

While the performance gains are apparent (modulo some experimental concerns discussed below), the significance of this approach may be somewhat limited, as it targets a specific application and offers incremental improvements, not new capabilities. Nonetheless, this will be of interest to those working on neural network verification.

The experimental analysis is limited, and focuses more on microbenchmarking LP solvers and SOL's isolated performance than showing gains in its most useful application, robustness verification.
1. Verification results are shown over two models, configured with three different activation functions. It's interesting to see the differences across activations, but this doesn't show consistent improvements as architectures grow deeper or wider. Additionally, MNIST results don't offer too many insights at this point, as prior work on certified training yields results than are difficult to improve on.
2. The experiments don't vary the robustness radius, so we can't determine how this impacts runtime or certification rates. In particular, it would be good to see that this approach's reliance on sampling doesn't break down for stronger guarantees. The 1/255 radius for CIFAR is not really standard anymore; although not a decisive argument, note that the [certified robustness community leaderboard](https://sokcertifiedrobustness.github.io/leaderboard/) doesn't have papers in this category, as most evaluate 2/255 or 8/255.
3. Important details of these models are not given: how many parameters, which layers, and how were they trained? Importantly, were these trained using techniques that would make them efficiently certifiable, or adversarial training, or something else? This will have a significant effect on the results. If these models were not trained for certification, then the experiments should include some models that were, as AutoLiRPA's default bound and LinSyn are likely to show improvement, and we would hope to see a comparable improvement in this approach.

The limited experiments make it difficult to judge the significance of this work, which is positioned as a drop-in replacement for existing bound approximation methods within verifiers.

**Questions:**

1. Please provide information on the missing details discussed in (3) from above.
2. If you have done experiments with different robustness radii or architectures, but did not report them, please describe the results.

**Limitations:**

Aside from the questions about experimental details discussed above, limitations were adequately addressed.

---

> ### Author Rebuttal · Authors · 2023-08-10
>
> Thank you for your review and the feedback.
>
> [exposition of the algorithm] Thank you for the suggestion. We will try to rearrange some of the paragraphs to make the narrative more coherent.
>
> [information on the missing details discussed in (3)]
> Here are the details. We will make them more clearly specified in the paper:
> 1) All of the models were trained by optimizing a simple cross-entropy. No robust training techniques were used. This is the setup
> LinSyn paper uses for their experiments, so we chose a similar approach.
> 2) The networks with "4l" in the name have 2 convolutional layers followed by 2 fully-connected layers, the "5l" networks – 3 conv + 2 fc.
> 3) The activation layer corresponding to the name of the model is present after each main layer except the last one.
>
> [experiments with different robustness radii] We conducted additional experiments running SOL on CIFAR with perturbation radii 2/255, 4/255 and 8/255.
> Similarly to the results for 1/255 presented in the paper the certification rates for any other radius do not depend on whether
> the optimality target is set to 1e-7, 1e-5 or 1e-3. This means that for bigger radii the saturation still happens somewhere above 1e-3.
> The certification rates drop significantly with increased radius as can be seen here
> ```
>       gelu, loglog, swish
> 1/225 0.31  0.69    0.37
> 2/225 0.    0.38    0.02
> 4/225 0.    0.08    0.
> 8/225 0.    0.03    0.
> ```
> Supposedly, the drop is so steep due to the models themselves not being very robust to begin with.
> The running times also increase with increased perturbation scale. The change is more pronounced for the small eps = 1e-7
> ```
>       gelu, loglog, swish
> 1/225 400s  160s    340s
> 8/225 700s  312s    630s
> ```
> then for the larger eps=1e-3
> ```
>       gelu, loglog, swish
> 1/225 150s  87s     150s
> 8/225 179s  90s     165s
> ```
>
> We hope to get the LinSyn results for the increased radii soon, so that we can make the comparison complete.
> However, so far nothing indicates that the comparison might be qualitatively different from the 1/255 comparison presented in the paper.

---

> > ### Author Response · Authors · 2023-08-14
> > **Increased perturbation radii results**
> >
> > We apologize for the delay. Here are the LinSyn results on CIFAR with increased perturbation radii.
> >
> > LinSyn certification rates:
> > ```
> >       gelu, loglog, swish
> > 1/225 0.31  0.69    0.35
> > 2/225 0.    0.38    0.02
> > 4/225 0.    0.08    0.
> > 8/225 0.    0.03    0.
> > ```
> > LinSyn runtimes:
> > ```
> >       gelu, loglog, swish
> > 1/225 582s  355s    546s
> > 2/225 576s  349s    562s
> > 4/225 586s  361s    563s
> > 8/225 599s  360s    574s
> > ```
> > The certification rates for SOL are indeed at least the same as those of LinSyn across the whole spectrum of perturbations.
> > Interestingly, the certification runtime does not increase much for LinSin between perturbations of 1/255 and 8/255.
> > However, it is still much higher than that of SOL with $\varepsilon = 10^{-3}$.
> >
> > To better analyze the saturation of certification rates as $\varepsilon \rightarrow 0$ we also evaluate SOL with $\varepsilon = 10^{-2}$ and achieve the following rates
> > ```
> >       gelu, loglog, swish
> > 1/225 0.25  0.62    0.32
> > 2/225 0.    0.33    0.
> > 4/225 0.    0.08    0.
> > 8/225 0.    0.03    0.
> > ```
> > The results suggest that for the perturbations of 1/255 and 2/255 the saturation occurs somewhere between $10^{-3}$ and $10^{-2}$, while for the perturbations of 4/255 and 8/255 the value of $\varepsilon = 10^{-2}$ is already saturated. Seemingly, the saturation $\varepsilon$ increases with perturbation scale which should be beneficial in practice.
> >
> > The LinSyn results also support this saturation dynamics hypothesis: the implicit optimality accuracy LinSyn has is not quite good enough for it to saturate the certification rates at 1/255 (SOL rate for the swish network is better), but is enough to saturate the rates for 2/255, 4/255 and 8/255.
> >
> > Finally, we would like to emphasize that the main purpose of the evaluation presented in the paper was to compare SOL to the two known function-agnostic bounding approaches.
> > The evaluation on a more diverse set of NNs trained using various techniques would surely enrich the analysis. However, since none of the approaches rely explicitly
> > on any specific model characteristic or the training algorithm, we argue that the presented results are decisive enough to be representative on their own. Especially
> > considering the following:
> > 1) we follow the experimental setup of LinSyn, which, supposedly, was chosen by the authors to reflect the performance of their approach as well as possible;
> > 2) LinSyn aims to produce bounds tight in the same $L_1$-distance sense as our optimal bounds, so it makes sense to attribute the difference in the robustness certification
> > performance to SOL being able to produce tighter bounds using less time (see fig. 7); this should stay true for any alternative experimental setup.

---

> > > ### Comment · Reviewer_GjZS · 2023-08-16
> > > **Rebuttal reply**
> > >
> > > Thank you for providing these clarifications, and additional data.
> > >
> > > It's somewhat puzzling why it makes sense to evaluate robustness certification methods on (very) non-robust models, as certified results aren't useful that often on them. To be clear, I am not faulting the authors for following the methodology established in the LinSyn paper, but this paper would provide more useful information to its readers if it included models that could be certified reasonably often on non-trivial radii.
> > >
> > > > SOL being able to produce tighter bounds using less time (see fig. 7); this should stay true for any alternative experimental setup
> > >
> > > I do not follow this claim. Doesn't this imply that SOL's relative performance is not affected by the particular geometry of the model? This claim needs supporting evidence.

---

> > > > ### Author Response · Authors · 2023-08-17
> > > >
> > > > [non-robust models]
> > > >
> > > > Let us provide more details on the specific problem setting we had in mind while developing SOL. Suppose that you are given a concrete NN model which you want to assess in terms of the certified robustness. You don't have any control over the model's architecture or the training process used. Practical examples of this might be
> > > > * Using certified robustness as a secondary metric while developing an NN solution. Both the architecture and the training routine are guided by the primary metric, so robustness considerations do not have direct influence on them.
> > > > * Estimating the certified robustness of a third-party NN solution for the purpose of analyzing the safety of a bigger system the solution is incorporated in.
> > > >
> > > > We expect that a non-robust model trained by a non-robust technique would be a natural, if not prevalent, target for the robustness certification in this setting.
> > > >
> > > > Importantly, we believe that the whole idea of function-agnostic linear bounding makes the most sense in a setting like this. Indeed, if you had control over the model training and your goal was to certify the highest robustness possible you would pick the activations (or other nonlinearities) which are easy to linearly bound or otherwise facilitate the certification. You would not need a general-purpose certification approach.
> > > >
> > > > [SOL's relative performance depending on the geometry of the model]
> > > >
> > > > We hypothesize that since SOL basically does the same thing as LinSyn (produces $L_1$-tight bounds) but better and faster it should always outperform LinSyn in terms of the robustness certification performance. How exactly the performance compares between the two does indeed depend on the choice of NN model. The difference in runtimes would, for example, depend on how much time does the activation function bounding take compared to the other processes in the certification pipeline.
> > > >
> > > > However, given the quite high 4x speedup measured in the experiments we conducted we find it likely that the performance improvement SOL delivers is noticeable for a large class of different neural networks and does not generally become negligible.

---

> > > > > ### Comment · Reviewer_GjZS · 2023-08-21
> > > > > **Thank you for your follow-up**
> > > > >
> > > > > > We expect that a non-robust model trained by a non-robust technique would be a natural, if not prevalent, target for the robustness certification in this setting.
> > > > >
> > > > > I remain skeptical about the usefulness of this, but I don't think that this matter should be decisive on the technical merit and significance of this work.
> > > > >
> > > > > > We hypothesize that since SOL basically does the same thing as LinSyn ...
> > > > >
> > > > > Thank you for providing these clarifying intuitions. I am somewhat more convinced, and am raising my score accordingly.

---

### Official Review · Reviewer_hjHu · 2023-07-19

**Soundness:** 4 excellent
**Presentation:** 4 excellent
**Contribution:** 4 excellent
**Rating:** 7
**Confidence:** 5

**Summary:**

[Context]
Neural network verification algorithms are based on bounding the activations and outputs of neural networks.
This is achieved by propagating linear bounds through the network.
For each non-linear operation in the network, it is required to provide linear bounds of the operation: given the range of inputs that the function accepts, give a linear mapping that upper (or lower) bound the function.
Using these linear bounds, propagation algorithm like the variants of Crown / Lirpa are able to bound the whole network.
These linear bounds of operations are usually manually derived.

[Contribution]
The authors define an optimality criterion in the context of convex or concave activation function [Theorem 1].
The main contribution of the paper is an algorithm to computationally derive linear bounds for functions that are not convex or concave, as long as they are lipschitz continuous. The algorithm consists in sampling a finite number of points, solving a LP to obtain the optimal linear bound that upper bound the sampled points [4.1] , and using smoothness criterion to adjust these bounds and guarantee that it will be a valid upper bound [4.2]. This procedure can be re-iterated until the desired accuracy gap (to the optimal linear bound) is achieved (4.3)


[Opinion of the paper]
I think that the paper is very interesting and the proposed algorithm can be very useful. The general idea of the algorithm is clearly presented, however some important details are missing which makes the current description of the paper insufficient to reproduce / re-implement it. If my main weaknesses comments ([Missing Description], [Mising Experiment])  and questions ([Computational aspects] [Details about the experiments]) are addressed, I would be happy to increase my Soundness to good/excellent and my General Rating to Weak Accept / Accept.


**Strengths:**

* The problem solved is important, and the proposed solution is interesting. Having had to derive manually convex relaxations of activation functions for some project, I definitely appreciate the value of an automated algorithm to handle this problem.

* The validation of the proposed measure is broad.
It is both performed on small problems where internal design choices (picking the LP solver algorithm) can be validated [5.1]  and compared to the most relevant baseline [5.2], but also on more (I assume, see Questions) large scale problems.

* The paper is very well structured, framing first the problem and their work in the context of the existing literature, introducing the framework in which they operate, and then building the description of their algorithm part by part in a logical fashion. While some elements are not fully described (see Weaknesses + Questions), I believe that this can be remedied for the final version.

**Weaknesses:**

[Definition of the Optimal Linear Bounding Problem]
The paper motivates the choice of the "Minimum volume criterion" by citing [36, 25, 13].
I could not find anything relevant to that point in the Popqorn paper [13]. In the DeepPoly paper [25] and Crown paper [36], the choice of lower bound for ReLU is indeed done by minimizing the area of the relaxation created, but without actually showing that it "correlates well with the performance of robustness certification" (except by showing that it performs by the old "parallel bounds"), and we know that those choices can be improved (as evidenced by paper like alpha crown that give better robustness results that those choices). In addition, no point is made about this for general functions beyond ReLU.
I completely agree with the authors that this is probably a good criterion, but if this is to be a basis for the method, I think the paper would be much stronger if this point was more strongly defended.
Some possible way this could be done:
- Can we link final robustness results to the volume criterion in some way? Maybe in expectation over some random weights, or at least clarifying that on a restricted type of activation function (convex? Monotonously increasing?), the volume optimal bound can be shown to be dominating other bounds.

[Missing description]
The "1D bisect" algorithm seems critical for the performance of the method (around twice as fast as the best alternative option), but it isn't described anywhere. The LP to solve need to return a slope vector $\mathbf{a}$ and an intercept $b$, but the only explanation given is in line 288. and 289 which does not describe how to implement those values.

[Missing experiment]
One experiment that is missing is the comparison to a network where we *know* what the correct bounds should be. If you take a network trained with sigmoid or tanh activation, the handcrafted convex hull / optimal linear bounds have been derived and should be tested (in the setting of Table 1). It is expected that the proposed SOL method would lose to that baseline, but knowing the gap in performance would be quite valuable, so that the "cost" of not deriving manually optimal bounds is known.


**Questions:**

[Results of Theorem 1]
This might be a result of the definition of "optimal", but in the case where conv_R(f) is not differentiable, but only subdifferentiable, then it is possible that we have several functions which are all "optimal" at the same time, is that correct?

[Claim in l.136 to l.142]
Unless I'm mistaken, both Softmax and SELU are neither convex nor concave, so should not be in the list of functions that are directly supported before introducing SOL.

[Claim in l.190 to l.192]
"It can always be chosen in such a way that $|a| \leq L_1$"
This claim doesn't seem justified by either a proof sketch or a reference?

[Computational aspects]
It's unclear to me on what plaftorm is every operation run? I assume that Gurobi and Scipy are running on CPU, but what about the other algorithms? Is the implementation of "1D bisect" amenable to vectorization?

[Details about the experiment]
The paper is missing some information about what type of networks are used for the "end to end" test including the method in AutoLIRPA. What activations were those network using? What size are they? How were they trained? Are they from a standard "NN robustness verification" benchmark such as the one from VNN-comp?

[Suggestion for future work - Ignore this for the review process]
It seems like this method could potentially be improved quite significantly by introducing some sort of smart caching. Most of a time, a network will use the same activation function through the network and it seems non optimal to restart from a uniform splitting of the domain every time, even if certain points (inflection points, local extrema) are much more likely to be important to include in the sample.

[Notes]
* Throughout the whole paper, Lipschitz is spelled Lipshitz. I did not find any examples of that spelling anywhere so I'm wondering if it's a typo or a different accepted spelling.

**Limitations:**

The paper would be improved by adding an experiment showing the cost it introduced vs. deriving hand crafted optimal bounds. See the [Missing experiment] comment in Weaknesses.

---

> ### Author Rebuttal · Authors · 2023-08-10
>
> Thank you for your review and the detailed feedback.
>
> [Missing description] We should definitely describe the 1d algorithm in more details. The idea of the bisection is as follows
> 1) guess the value $g(x_c) = y$ which we want to optimize;
> 2) check its feasibility by finding $a = \min_{x_i < x_c} \frac{y - f(x_i)}{x_c - x_i}$ and $b = \max_{x_i > x_c} \frac{f(x_i) - y}{x_i - x_c}$ in $O(|S|)$ time;
> 3) if $a < b$, the guess is infeasible since to line passing $(x_c, y)$ upper-bounds both argmaxes from the previous step;
> 4) otherwise, the guess is feasible and a valid solution would be $g(x) = y + (x - x_c) (a + b) / 2$.
>
> So, starting with the initial feasible guess of $\max f(x_i)$ and the initial infeasible guess of $\min f(x_i)$ we can find the solution within $\varepsilon$ of optimum in $O(|S| \log \frac{1}{\varepsilon})$ time.
>
> [Missing experiment] Such an experiment would, indeed, be a great addition to the paper and we will gladly include it.
> Right now similar quantities can be estimated by comparing:
> 1) Certification rates of SOL with higher eps and the rates of SOL with low enough eps for rates to be saturated (1e-7).
> The saturated rates should be close enough to the hypothetical rates of returning the exact optimal bounds.
> 2) Running times of SOL and the running times of the default AutoLIRPA. For simpler functions decompositional bounding of AutoLIRPA
> should take time similar to how much calculating optimal hand-crafted bounds takes.
> Higher times for gelu networks are supposedly caused by massive decompositional structure of the function.
>
> [Definition of the Optimal Linear Bounding Problem] Connecting the choice of the tightness measure to the robustness properties would be very appropriate for the paper.
> We will look into it. Right now the motivation is more practical:
> 1) other researchers have tried it and didn't find it pathological;
> 2) this exact tightness measure is convenient to optimize.
>
> Also, we believe that for many practical scenarios the exact choice of a measure (withing a certain spectrum) might no affect the optimal bound much.
>
> [Results of Theorem 1] Correct, any subderivative passing through $(x_c, conv_R(f)(x_c))$ would be optimal.
> E.g., optimal bounds for $-|x|$ in [-1, 1].
>
> [Claim in l.136 to l.142] For SELU we meant to refer to the case when the scale parameter of the linear portion of the function is set high enough for the function to be convex. Softmax, indeed, should not be mentioned there. What should have been mentioned instead is max_pool – which is a convex function of several variables.
>
> [Claim in l.190 to l.192] Thank you for noticing this one! A few weeks ago we gave this statement a little more thought and realized that it's not as straightforward as it had seemed.
> 1) Given an arbitrary solution to the discrete problem we can always "perturb" it in such a way that the corresponding hyperplane passes through $d + 1$ points $[(x_i, f(x_i))]$ of the sample and the center of mass $x_c$ lies in the closed simplex $[x_i]$.
> 2) For the one-dimensional case this is enough to satisfy the statement: $|a| = \frac{|f(x_1) - f(x_2)|}{|x_1 - x_2|} \le L_1$.
> 3) In higher dimensions it does not necessarily hold as is. Note that for an arbitrary point $x \in S$ the upper-bounding condition requires
>        $$f(x_i) + a(x - x_i) \ge f(x) \ge f(x_i) - L_1 |x - x_i|$$
> therefore
>         $$a \frac{x - x_i}{|x - x_i|} \ge - L_1 \Rightarrow (-a) \frac{x - x_i}{|x - x_i|} \le L_1 \Rightarrow |a| \le L_1 / (\frac{-a}{|a|} \cdot \frac{x - x_i}{|x - x_i|}).$$
> This gives us the general bound of
>         $$|a| \le L_1 / \inf_{[x_i], v} \sup_{i, x \in S} (v \cdot \frac{x - x_i}{|x - x_i|}),$$
> where infimum is taken over all simplexes $[x_i]$ such that $x_c$ lies in the simplex and $v$ are all possible unitary vectors corresponding to the direction of the gradient.
> It's easy to see that that this min-max optimization depends on two things:
> 1) the density of points in the sample: higher density gives lower bound;
> 2) the geometry of R which determines the limit value of the bound when density approaches infinity
>             $$q =  \inf_{[x_i]\subset R, v} \sup_{i, x \in R} (v \cdot \frac{x - x_i}{|x - x_i|}).$$
> The geometric factor $q$ is quite important in practice. Intuitively, $R$ having "acute angles" on the boundary may make $q$ smaller than 1.
>
> One example of such situation is in 2d: let $R$ be the triangle {(0, 0), ($w$, 1), (-$w$, 1)}, for which $q \rightarrow 0$ as $w \rightarrow \infty$. If we chose the target function as $f(x,y) = |x|$, the Lipschitz constant is always $L_1 = 1$, but the optimal upper bound is $g(x,y) = wy$ with $|a| = w \rightarrow \infty$.
> And the estimated discrete optimal bounds might have similarly big |a|.
>
> On the other hand, $R$ with "no angles" – a closed sphere seems to have $q = 1$ in any dimension, so we can guarantee $|a| <= L_1 (1 + \varepsilon)$ for any positive $\varepsilon$ if the initial sample size is big enough.
>
> All in all, these factors only introduce an additional constant factor into the complexity.
>
>
> [Computational aspects] All methods only use CPU. We run the experiments from the paper in a VM with 3 cpus allocated.
> AutoLIRPA runs several instances of the bounding problem solver concurrently.
> The calculation of two $\frac{f(x_i) - y}{x_i - x_c}$ arrays in 1d bisect algorithm should be vectorizable. We will look into it.
>
>
> [Details about the experiment]
> These details should definitely be included in the paper:
> 1) We run AutoLIRPA with the simple "CROWN" method for bounding.
> 2) The networks with "4l" in the name have 2 convolutional layers followed by 2 fully-connected layers, the "5l" networks – 3 conv + 2 fc.
> 3) The activation layer corresponding to the name of the model is present after each main layer except the last one.
> 4) All of the networks are trained by optimizing traditional cross-entropy. Similarly to how it was done in LinSyn paper.
>
> [Lipschitz vs Lipshitz] It seems to be a typo in the spell-checker on our side.

---

> > ### Comment · Reviewer_hjHu · 2023-08-11
> >
> > **Re:[Missing description]**
> > Thanks for the explanation. Does it apply as well to the case where x is a vector, rather than a scalar? Or is the method limited to 1d activation function? (This is fine if that is the case, but it should be pointed out).
> >
> > Thank you for the detailed response. I have increased my scores.

---

> > > ### Author Response · Authors · 2023-08-14
> > >
> > > This version of the bisect procedure only applies to the 1d case. We haven't been able to generalize it to the higher dimensions yet.
> > >
> > > We used to have a couple of words mentioning this limitation, but seem to have lost them along the way. The "1D" in "1D bisect" was meant to highlight the fact.
> > >
> > > Again, thank you for noticing these details!

---

### Official Review · Reviewer_173q · 2023-07-20

**Soundness:** 3 good
**Presentation:** 3 good
**Contribution:** 2 fair
**Rating:** 4
**Confidence:** 4

**Summary:**

This paper focuses on finding tight linear bounds for the activation functions in neural networks, which are used for certifying the robustness of a neural network.

It aims to provide optimality guarantees for the tightness of the bounds of a scalar function represented by the neural net.
Two settings are considered:
1. for functions that are convex in some region $R$, optimal bounds are obtained through an optimality criterion for the tightness of the approximation in $R$;
2. for functions that are Lipschitz continuous in some region $R'$, a sampling-based approach is proposed called SOL. Given an instance of the bounding problem and a positive scalar $\epsilon$, SOL efficiently computes the tightest linear bounds within the $\epsilon$ threshold.

The empirical simulations show that the proposed method SOL typically takes a quarter of the time other methods take.




**Strengths:**

**Novelty to more systematically analyze the linear bound tightness.**
Although I am not working in this area, it seems this paper is among the first to study the tightness of the linear bounds for the activation functions used for neural nets.

**Elegant/Simple approach for the convex setting.**
The results for convex functions are very simple, but to me, that makes the approach very nice: it shows that it suffices to estimate the center of the mass, which is a point in the given region from the domain (in practice estimated after sampling), and compute the gradient at that point to obtain a linear bound.




**Weaknesses:**

**Incomplete results for $L$-Lipschitz setting.**
While I personally enjoyed the overall idea, I think the results for the $L$-Lipschitz setting are incomplete.
In particular:
- these should be more rigorous (see also comment below), and moreover
- I expected to see a theorem that relates the tightness of the bounds estimated with sampling with the $L$-constant, the sample size, the dimensionality of the problem, etc. (even if this is done for some specific non-convex function).


**Writing.**
The writing is, in general, easy to follow.
However, parts of the text are relatively ambiguous or less formal.
For example, the following should be improved:
- *relating to motivation*: since robustness certificates seem to be the main motivation mentioned in the paper for linear bounds, explain how the latter gives the former (and if it is the most consuming step)
- *Prop. 1*. The way it is written, the proposition is not a full statement -- write it more formally to include the previous definition of the volume or point to it when making the statement.
- *Theorem 1*. Similarly, theorem 1, as stated, is incomplete -- the setting and the assumptions should be repeated here.
- *Missing proof of Thm. 1*.
- *the discussions for the $L$-Lipschitz setting should be made much more rigorous.* (and stated as lemmas/theorems).
- Fig. $3$ should be explained better in the caption.
- etc.




## Minor

- Abstract: *tightness optimality* is hard to understand; elaborate better as it seems central
- line 157: *the* sample $S$

**Questions:**

1. What is the average time complexity of SOL, and does it relate better with the observed time? If not, could you discuss what are the limitations / worst-case examples that yield the worst-case time complexity?


**Limitations:**

Please see above.

---

> ### Author Rebuttal · Authors · 2023-08-10
>
> Thank you for such a detailed feedback. We will definitely address clumsiness in formulations and unintentional ambiguities.
>
> [average time complexity of SOL] Unfortunately, we cannot provide a rigorous analysis of the average time complexity, since the dynamics of the estimated linear bound
> between the iterations seems too complex. However, the proposed intuitive reasoning (lines 252-261) aligns pretty well with what we get in practice (fig. 5a, 5c).
> The empirical complexity of $O(\log^2 \frac{1}{\varepsilon})$ can be attributed to:
> 1) 1d bisect taking $O(n \log \frac{1}{\varepsilon})$ time to solve each discrete problem with $n$ points;
> 2) the final discrete problem typically having $O(\log \frac{1}{\varepsilon})$ samples concentrated near the touching point as per the reasoning;
> 3) the number of points growing fast enough for the last iteration's runtime to be dominant.
>
> [worst-case examples] Bounding linear functions gives complexities close to our worst-case bound of $O(\varepsilon^{-2d})$ (lines 246-251). Currently, we don't know whether closer
> complexities are achievable on not.
>
> [theorem that relates the tightness of the bounds ...] Such a relation is given implicitly for the uniform (not adaptive) SOL at line 227 by defining the cell size
> in terms of the desired optimality and the Lipschitz constant. We should definitely make this relation more explicit for the sake of clarity.
> For the adaptive version, again, no rigorous connection of this sort is known to us.

---

### Official Review · Reviewer_Uxf6 · 2023-07-27

**Soundness:** 3 good
**Presentation:** 3 good
**Contribution:** 3 good
**Rating:** 7
**Confidence:** 3

**Summary:**

This paper introduces a new method for finding tight (in the l1 error sense) linear bounds for a general non-linear activation functions. Under their definition of tightness, the authors propose a method for finding optimal bounds for any convex function. For non-convex functions with Lipschitz continuity, they propose a sampling based method that efficiently computes the linear bounds within some error threshold. The performance of the proposed methods are benchmarked in robust certification tasks.

**Strengths:**

This is an interesting paper tackling an important problem (linear bounding of arbitrary activation functions) in the field. It formulates and solves the problem in a rigorous fashion, and a systematic benchmarking proves the utility of the new method.

**Weaknesses:**

I do not see particular weaknesses in this paper, but I do have a question out of theoretical interest. The optimality criterion is based on the L1 error of between the bounds g(x) and the activation function f(x). And this L1 error is important in the derivation of Proposition 1. I wonder how the landscape would change if we change the loss function to, e.g., L2 or other meaningful functions.

**Questions:**

Please see Weakness section above.

**Limitations:**

I do not see any limitations.

---

> ### Author Rebuttal · Authors · 2023-08-10
>
> Thank you for your review and for the interesting question.
>
> [significance of optimizing $L_1$ error] Right now we don't see any straightforward way to generalize our approaches to alternative tightness measures.
> Moving from $L_1$ measure to, for example, $L_2$ poses several immediate problems:
> 1) As you have already mentioned, Proposition 1 no longer works. What is worse, now the loss function depends on $\int_R x g(x)$ and $\int_R g(x)$ in a non trivial way.
> Generally, there is no analytical expression for these two.
> 2) The analogue of our discrete problem is no longer an LP, but a QP, which is harder.
> 3) Shifting the bound up to adjust for possible unsoundness of the discrete problem's solution no longer corresponds directly to incrementing the loss function.
>
> On the other hand, we expect that, at least in 1d, in many situations (when the optimal $L_1$ bound touches $f(x)$ in two distinct points) the optimal bound stays the same for
> a wide spectrum of discrepancy measures. Therefore, the exact choice of measure may not influence the performance of the robustness certification too much.
> Then we might as well choose a measure which facilitates the optimization.

---

> > ### Comment · Reviewer_Uxf6 · 2023-08-13
> >
> > I would like to thank the authors for the clarification of my question. And I will keep my score.

---

### Decision · Program_Chairs · 2023-09-21

**Decision:**

Accept (poster)

**Comment:**

This paper introduces a technique for estimating linear bounds on neural networks for use in certification settings. The approach is able to deal with arbitrary lipschitz activation functions without requiring manual user construction of bounds, thus suggesting a path towards automation of the procedure. The reviewers recognize the utility of this result and recommend acceptance.